# Effect of Combined Surgery in Patients with Complex Nanophthalmos

**DOI:** 10.3390/jcm11195909

**Published:** 2022-10-07

**Authors:** Yantao Wei, Yihua Su, Lei Fang, Xinxing Guo, Stephanie Chen, Ying Han, Yingting Zhu, Bing Cheng, Shufen Lin, Yimin Zhong, Xing Liu

**Affiliations:** 1State Key Laboratory of Ophthalmology, Zhongshan Ophthalmic Center, Sun Yat-sen University, Guangdong Provincial Key Laboratory of Ophthalmology and Visual Science, Guangdong Provincial Clinical Research Center for Ocular Diseases, Guangzhou 510060, China; 2The Ophthalmology Department, The First Affiliated Hospital of Sun Yat-sen University, Guangzhou 510060, China; 3Wilmer Eye Institute, Johns Hopkins University, Baltimore, MD 21287, USA; 4Department of Ophthalmology, San Francisco School of Medicine, University of California, San Francisco, CA 94143, USA

**Keywords:** nanophthalmos, glaucoma, vitrectomy, phacoemulsification, posterior capsulotomy

## Abstract

(1) Background: To evaluate the efficacy and safety of combined surgery (limited pars plana vitrectomy, anterior-chamber stabilized phacoemulsification, IOL implantation and posterior capsulotomy, LPPV + ACSP + IOL + PC) in complex nanophthalmos. (2) Methods: Patients with complex nanophthalmos were recruited to undergo LPPV + ACSP + IOL + PC from January 2017 to February 2021. Preoperative and post-operative intraocular pressure (IOP), best corrected visual acuity (BCVA), anterior chamber depth (ACD), and number of glaucoma medications were compared using the paired t-test or Wilcoxon signed rank sum tests. Surgical success rate was evaluated. Surgery-associated complications were documented. (3) Results: Forty-five eyes of 37 patients with complex nanophthalmos were enrolled. The mean follow-up period was 21.7 ± 10.6 months after surgery. Mean IOP decreased from 32.7 ± 8.7 mmHg before surgery to 16.9 ± 4.5 mmHg (*p* < 0.001) at the final follow-up visit, mean logMAR BCVA improved from 1.28 ± 0.64 to 0.96 ± 0.44 (*p* < 0.001), mean ACD significantly increased from 1.14 ± 0.51 mm to 3.07 ± 0.66 mm (*p* < 0.001), and the median number of glaucoma medications dropped from 3 (1, 4) to 2 (0, 4) (*p* < 0.001). The success rate was 88.9% (40 eyes) at the final follow-up visit. Two eyes had localized choroidal detachments which resolved with medical treatment. (4) Conclusions: LPPV + ACSP + IOL + PC is a safe and effective surgical procedure, which can decrease IOP, improve BCVA, deepen the anterior chamber, and reduce the number of glaucoma medications in patients with complex nanophthalmos. It can be considered as one of the first treatment in nanophthalmic eyes with complex conditions.

## 1. Introduction

Nanophthalmos is a relatively rare disease characterized by a short axial length (AL, <20.0 mm), crowded anterior chamber, high lens/eye volume ratio, and axial hypermetropia [1,2,3,4]. Angle-closure glaucoma (ACG) is a common complication in eyes with nanophthalmos, mainly due to pupillary block, displacement of the peripheral iris, and development of peripheral anterior synechia (PAS) [5,6]. The treatment in nanophthalmic eyes, especially in complex nanophthalmos with acute angle-closure glaucoma (AACG) or with failed glaucoma surgery, is difficult and challenging. Usually, the response to medical treatment is poor in such patients. Prior studies supported performing laser or surgical peripheral iridectomy to control intraocular pressure (IOP) in the early stages of nanophthalmos-induced ACG [7,8,9,10,11]. However, surgical intervention is usually required in more severe disease with uncontrolled IOP or progressive shallowing of the anterior chamber. Due to the multiple anatomic abnormalities in nanophthalmos eyes, routine operations become quite challenging and associated with high incidences of significant intra- and postoperative complications, such as malignant glaucoma, uveal effusion, and uveal hemorrhage, as previously reported [12,13].

Phacoemulsification surgery has been shown to be a feasible option for patients who are diagnosed with microphthalmos or nanophthalmos and cataracts and desire to improve vision and are willing to accept the risks of surgical complications [14,15,16]. For nanophthalmic eyes with glaucoma, disproportionately large lenses are thought to contribute to pupillary or ciliolenticular block in the pathogenesis of glaucoma [5]. Theoretically, phacoemulsification alone can help to eliminate lens-associated block and reconstruct the aqueous outflow pathway, thereby treating complex nanophthalmos. Nevertheless, previous assessments of intraocular surgery in nanophthalmic eyes, such as phacoemulsification alone, documented high incidences of uveal effusion or intraocular hemorrhage, primarily due to intraoperative rapid IOP fluctuations and sudden shallows in anterior chamber depth (ACD) when removing the intraocular instruments (phaco or irrigation-aspiration (I/A) tips) during an operation [17,18,19]. Thus, maintenance of IOP and ACD were considered critical throughout the entire operation. In this study, we describe a combined surgical technique for stabilizing IOP and anterior chamber during cataract surgery for complex nanophthalmos. This technique involves limited pars plana vitrectomy, anterior-chamber stabilized phacoemulsification, IOL implantation and posterior capsulotomy (LPPV + ACSP + IOL + PC). Moreover, the efficacy and safety profile of this combined procedure are evaluated.

## 2. Materials and Methods

This retrospective case series was performed in accordance with the World Medical Association’s Declaration of Helsinki and was approved by the Ethics Committee of the Zhongshan Ophthalmic Center, Sun Yat-sen University (2021KYPJ181). Informed consent was obtained from each subject or their guardians. 

### 2.1. Patients

This study included patients with complex nanophthalmos between January 2017 to February 2021. The inclusion criteria for our study were as follows: (1) diagnoses of nanophthalmos; (2) at least 12 months of follow-up after surgery; (3) combined with complex conditions, as evaluated by the attending physician meeting, including at least one of the following clinical criteria: (i) secondary AACG; (ii) with uncontrolled IOP or had complications (such as malignant glaucoma and extremely shallow ACD (<1 mm)) following a previous failed glaucoma surgery. The previous failed glaucoma surgeries included laser or surgical peripheral iridectomy (LPI, SPI), trabeculectomy, or ultrasound cycloplasty (UCP). Patients with uncontrolled ocular infection and severe systemic diseases were excluded. Nanophthalmos was defined as having a shorter AL (<20.0 mm), a shallow anterior chamber, high lens/eye volume ratio, moderate to severe hyperopia, and choroidal-scleral thickening measured by B-scan ultrasonography (Quantel Medical, CF, Cournon d’Auvergne Cedex, France) and optical coherence tomography (OCT, Heidelberg Engineering, Heidelberg, Germany) [20,21,22]. The medical records of the enrolled patients were recorded.

### 2.2. Examinations

All patients underwent ophthalmic examinations, involving best-corrected visual acuity (BCVA) testing with Snellen charts, IOP measurements with Goldmann applanation tonometry, refraction, slit-lamp biomicroscopy, and ophthalmoscopy. For subjects with clear cornea, gonioscopy was performed by an experienced glaucoma specialist (X.L.) to evaluate the degree of angle closure. The ACD was measured by anterior segment of optical coherence tomography (AS-OCT, CASIA SS-1000TM, Tomey, Aichi, Japan). Ciliary body and zonule was evaluated by ultrasound biomicroscopy (UBM, model SW-3200L; Tianjin Sower Electronic Technology Co., Ltd., Tianjin, China). Axial length (AL) and lens thickness (LT) were measured by A-scan ultrasound biometry (Quantel Medical, CF, France). B-scan ultrasonography was performed to assess for retinal detachment or uveal effusion. The horizontal corneal diameter (HCD) was measured by a keratometer (WAM-5500 Grand Seiko, Hiroshima, Japan). Visual field (VF) testing was performed by standard automated perimetry with the SITA standard 30–2 program (Zeiss Humphrey visual field 750i, Carl Zeiss Meditec Inc., Dublin, CA, USA). Patients with BCVA <20/200 or poor fixation did not undergo VF testing. Color fundus photography (TRC-NW6S; Topcon, Tokyo, Japan), scanning laser ophthalmoscopy (Optos PLC, Dunfermline, UK), and OCT were performed if permissible based on media clarity. 

### 2.3. Surgical Procedure

One hour before surgery, 20% mannitol was intravenously administered to reduce intraoperative vitreous pressure in all cases. All surgeries were performed under general anesthesia by two experienced ophthalmologists (X.L, Y.W). 

#### 2.3.1. Limited Pars Plana Vitrectomy (LPPV)

A 25-gauge trocar was placed in a transconjunctival and transscleral fashion in the inferotemporal quadrant 2 mm posterior to the limbus in order to account for the extremely short AL (Figure 1A,B). With the 25-gauge Constellation system (Alcon Laboratories, Fort Worth, TX, USA), a limited pars plana vitrectomy was performed via this trocar to moderately reduce posterior pressure and deepen the anterior chamber. The vitrectomy cut rate was set at 5000 cpm to minimize vitreous traction. 

#### 2.3.2. Anterior-Chamber Stabilized Phacoemulsification and IOL Implantation (ACSP + IOL)

A 3.2 mm temporal clear corneal incision was created, followed by injection of a viscoelastic agent (Healon 5, Abbott Medical Optics, AMO, Santa Ana, CA, USA) to sufficiently flatten the iris and deepen the anterior chamber (Figure 1C). Two paracenteses were made at 12 and 6 o’clock in the left eye and 11 and 1 o’clock in the right eye. Posterior synechiolysis or sphincterotomies were performed as needed to facilitate pupillary dilation and maximize visualization. A 5.5 mm continuous curvilinear capsulotomy was completed with Utrata forceps. Phacoemulsification of the nucleus and I/A of the cortex were performed. To help prevent IOP fluctuations and stabilize the anterior chamber, prior to removal of the intraocular instruments (phaco or I/A tips), the viscoelastic agent was injected into the anterior chamber from one of the paracenteses (Figure 1D,E). In all patients, a foldable, single-piece intraocular lens (IOL) (Acrysof SA60AT; Alcon) was implanted in the capsular bag (Figure 1F). 

#### 2.3.3. Posterior Capsulotomy (PC) 

A 4–5 mm posterior capsulectomy (capsulotomy) was performed with the 25-gauge vitrector (Figure 1G). Once completed, the trocar was removed. The incision site was sealed without suturing (Figure 1H). The remaining viscoelastic agent in the anterior chamber was then carefully exchanged with balanced salt solution.

Postoperatively, prednisolone acetate 1% (Allergan, Parsippany-Troy Hills, NJ, USA) and topical antibiotics (Ofloxacin, Santen Pharmaceutical Co., Ltd. Noto Plant, Osaka, Japan) were administered four times daily for 4 weeks. All patients were followed at postoperative day 1, week 1, month 1, month 3, and every 3 months thereafter. Glaucoma eye drops or oral medications were administered for IOP > 21 mmHg. 

The occurrence of intraoperative complications, including dropped nucleus, iris prolapse, anterior chamber hemorrhage, choroidal detachment, uveal effusion, or suprachoroidal hemorrhage, were noted. Postoperative complications were also documented, including uveal effusion, choroidal detachment, vitreous hemorrhage, malignant glaucoma, rhegmatogenous or serous retinal detachment, and persistent iritis, as was the need for additional surgeries during the entire follow-up period.

Surgical success was defined as: (1) postoperative IOP (≥5 mmHg and ≤21 mmHg) with or without use of glaucoma medications; (2) the absence of severe, vision-threatening complications, such as suprachoroidal hemorrhage, retinal detachment, endophthalmitis, or loss of light perception; and (3) not requiring additional glaucoma surgery.

### 2.4. Statistical Analysis

For statistical analyses, BCVA was converted to logarithm of the minimum angle of resolution (logMAR) visual acuity. Counting fingers, hand motion, light perception, or no light perception vision were noted as logMAR values of 2.1, 2.4, 2.7, and 3.0, respectively [23]. Descriptive statistics were reported as means and standard deviations (SD), medians and interquartile ranges, or numbers and percentages as appropriated. The normality of continuous variable distributions was examined using the Kolmogorov–Smirnov test. Paired t-tests and Wilcoxon signed rank sum tests were used to assess differences between preoperative and postoperative values according to whether variables conformed to a normal distribution. Statistical significance was defined as *p* < 0.05. All statistical analyses were performed using SPSS software, version 22.0 (SPSS, Inc., Chicago, IL, USA).

## 3. Results

### 3.1. Patient Characteristics

LPPV + ACSP + IOL + PC surgery was performed on 45 eyes of 37 patients with complex nanophthalmos. There were 12 (32.4%) male and 25 (67.6%) female patients. The mean age was 46.6 ± 12.2 years (range 17–78 years). Thirteen eyes (28.9%) presented secondary AACG with uncontrolled IOP despite maximum tolerated medical therapy. Thirty-two eyes (71.1%) had undergone prior glaucoma procedures, including LPI in 12 eyes, SPI in 13 eyes, trabeculectomy in 5 eyes, and UCP in 2 eyes. Of the 32 eyes, 28 eyes had uncontrolled IOP following previous surgery. Among them, 11 eyes developed malignant glaucoma. The remaining four eyes had extremely shallow ACD (<1 mm) after surgery. The clinical parameters at baseline are summarized in Table 1. The mean duration of follow-up after surgery was 21.7± 10.6 months (range 15–72 months).

### 3.2. Intraocular Pressure (IOP) and Surgical Success Rate

Preoperatively, the mean IOP was 32.7 ± 8.7 mmHg in our case series. Mean IOPs with or without glaucoma medications decreased to 17.2 ± 6.5 mmHg at 1 week, 18.6 ± 7.2 mmHg at 1 month, 18.4 ± 7.7 mmHg at 3 months, 19.5 ± 8.0 mmHg at 6 months, 19.1 ± 6.9 mmHg at 12 months and 16.9 ± 4.5 mmHg at the final follow-up visit postoperatively (Figure 2). The difference in IOP before and after surgery was statistically significant at all visits (all *p* < 0.001). 

Surgical success was found in 40 of 45 eyes (88.9%) at the last documented follow-up visit. Among these, 17 eyes (42.5%) did not require glaucoma medications. Five eyes (11.1%) were classified as failures because of the need for additional glaucoma procedures to adequately control IOP. Cyclophotocoagulation was performed in 2 of the 5 eyes, while the remaining 3 eyes underwent uncomplicated Ahmed glaucoma valve implantation. All 5 eyes achieved IOP control during the postoperative follow-up period (7.0 ± 4.6 months), with a mean final IOP of 11.4 ± 3.2 mmHg.

### 3.3. Best-Corrected Visual Acuity (BCVA)

At the final follow-up visit, the mean logMAR BCVA (0.96 ± 0.44) was significantly improved from baseline (1.28 ± 0.64) (*p* < 0.001). Of the 45 eyes, the BCVA of 33 eyes (73.3%) showed improvement, 10 eyes (22.3%) remained unchanged from baseline, and 2 (4.4%) eyes decreased (one eye lost 3 Snellen lines and 1 Snellen line in the other) but no eye lost vision. The two eyes had a decrease in visual acuity because of perioperative choroidal detachment. 

### 3.4. Anterior Chamber Depth (ACD) and Degrees of Angle Closure

Preoperatively, the mean ACD was 1.14 ± 0.51 mm. At the final visit after surgery, the mean ACD was significantly deeper at 3.07 ± 0.66 mm (*p* < 0.001). Gonioscopy was performed in 21 patients (25 eyes) before surgery. The median postoperative degree of angle closure reduced from 330° (range 0–360°) at baseline to 240° (0–360)° at the final visit; however, the difference was not statistically significant (*p* = 0.172).

### 3.5. Number of Glaucoma Medications

The median number of glaucoma medications was 3 (range 1–4) before surgery and significantly decreased to 2 (range 0–4, *p* < 0.001) at the final postoperative visit. A total of 17 eyes (17/40, 42.5%) did not require any glaucoma medication at the final follow-up. 

### 3.6. Surgical Complications

Intra- and postoperative complications were observed in 2 eyes (4.4%). A localized choroidal detachment was noted intraoperatively in one eye and on the first postoperative day in the other eye. In both, the choroidal detachment resolved with conservative treatment within 2 weeks. During the postoperative follow-up period, there was no sight-threatening postoperative complication, such as retinal detachment, uveal hemorrhage, hypotony, or endophthalmitis.

### 3.7. Typical Case

A 39-year-old male was referred to the glaucoma division in July 2019 with complaints of poor vision in both eyes since childhood and aggravation of the right eye (RE) associated with headaches for the past 6 months. He was subsequently diagnosed with nanophthalmos and secondary glaucoma in both eyes. Although LPIs had been performed in both eyes, the IOP in the RE could not be controlled with glaucoma medications. 

At his examination in January 2020, BCVA was 20/100 with +12.75 diopters of hyperopia RE and 20/125 with +13.00 diopters of hyperopia in the left eye (LE). IOP was 38 mmHg RE and 21 mmHg LE with brinzolamide (S. A. ALCON-COUVREURN. V, UK) and brimonidine tartrate eye drops (Allergan Pharmaceuticals, Ireland). Slit-lamp biomicroscopy showed shallow anterior chambers and mild nuclear sclerotic cataracts in both eyes (Figure 3A,B). The vertical C/D ratio was 0.5 RE and 0.2 LE (Figure 3C). A-scan ultrasonography revealed an AL of 16.19 mm RE and 16.48 mm LE. OCT demonstrated that the average retinal nerve fiber layer thickness was 87 μm RE and 95 μm LE, and the subfoveal choroidal thickness was 524 μm RE and 576 μm LE (Figure 3D). AS-OCT showed a central ACD of 0.75 mm RE and 0.76 mm LE.

The patient subsequently underwent bilateral LPPV + ACSP + IOL + PC successively. On the first day after surgery, his uncorrected visual acuity was 20/200 RE and 20/500 LE. The central ACD significantly increased (RE: 3.71 mm, LE: 3.33 mm) compared to the preoperative ACD (Figure 4A,B). B-scan ultrasonography and SLO showed flat and attached retinas in both eyes (Figure 4C,D). At the last visit (RE: 6 months after surgery, LE: 3 months after surgery), BCVAs of both eyes were 20/100. IOP was 21 mmHg RE with glaucoma medications (brinzolamide and brimonidine tartrate eye drops) and 20 mmHg LE without medication. 

## 4. Discussion

This study showed that limited pars plana vitrectomy, anterior-chamber stabilized phacoemulsification, IOL implantation, and posterior capsulotomy (LPPV + ACSP + IOL + PC) can effectively decrease IOP, deepen the anterior chamber, reduce the number of IOP-lowering medications, and improve BCVA with few complications in patients with complex nanophthalmos. 

Cataract surgery in nanophthalmic patients has always been technically challenging due to the limited space in the anterior chamber [14,15,16]. Previous studies showed that shorter AL and smaller ACD were associated with higher risks of complications [24,25]. In our enrolled subjects, the mean ACD was extremely shallow, and the mean AL was extremely short, even more so than those reported by Day [24] and Ye et al. [25]. Moreover, eyes with nanophthalmic glaucoma are usually associated with elevated vitreous pressure, which makes cataract surgery complicated and may lead to several secondary complications [26]. Therefore, limited PPV combined with phacoemulsification is a reasonable approach to such a high-risk situation. Sharma et al. [27] described combining PPV with phacoemulsification to successfully manage malignant glaucoma in phakic eyes. Using a 25-gauge system, Chalam et al. [26] also reported the advantages of limited vitrectomy to deepen the anterior chamber and facilitate phacoemulsification in eyes with positive vitreous pressure and shallow anterior chambers. However, combined PPV and phacoemulsification, as was previously performed for nanophthalmic glaucoma, has a few challenges. First, severe vision-threatening complications such as uveal effusion and uveal hemorrhage have been reported because of rapid IOP fluctuations and a sudden shallower anterior chamber during surgery [17,18,19]. Second, nanophthalmic patients at baseline have a high incidence of postoperative aqueous misdirection [28].

In this study, we described a combined procedure to manage eyes with complex nanophthalmos. In our surgical procedure, LPPV was considered the first step to reduce vitreous pressure and deepen the anterior chamber. A viscoelastic agent was used to strictly control the IOP and stabilize the anterior chamber throughout the entire surgery. Finally, a posterior capsulotomy was performed to reduce the risk of postoperative aqueous misdirection. Our results show that LPPV + ACSP + IOL + PC is effective and safe in the management of patients with complex nanophthalmos with minimal intra- and post-operative complications.

A significant reduction of IOP and deepening of ACD in eyes with complex nanophthalmos treated with LPPV + ACSP + IOL + PC was detected in the follow-up period, with an 88.9% success rate. The success rate of antiglaucoma surgery for nanophthalmic glaucoma varies in previous studies [1,11,29]. Singh et al. [1] reported that of 15 patients with nanophthalmic glaucoma undergoing filtration surgery, 60.0% failed treatment, and 86.6% suffered a visual loss. Yalvac et al. [11] reported that the total success rate was 85.0, 78.5, and 47.0% at 1, 2, and 5 years after trabeculectomy for patients with nanophthalmic glaucoma. Zhang et al. [29] reported results of 23-G vitrectomy combined with lensectomy in 21 eyes with nanophthalmic glaucoma. The total success rates were 85.7%, 81.0%, and 85.7% at the 6-month, 12-month, and final follow-up visits, respectively, consistent with our results. However, all the eyes in their study underwent lensectomy without IOL implantation, which resulted in the limited visual outcomes.

Generally, visual impairment in nanophthalmic patients is associated with amblyopia, macular abnormalities, or other ocular disorders [1]. In our cohort, poor vision before surgery was further exacerbated by glaucoma-related damage. The preoperative mean logMAR BCVA noted here (1.28 ± 0.64) was worse than that previously seen [11,30]. Yalvac et al. [11] reported a mean preoperative BCVA of 0.24 ± 0.15 in 20 patients (28 eyes) with nanophthalmic glaucoma, while Steijins et al. [30] described a median preoperative BCVA of 20/60 (HM to 20/25) or approximately 0.48 logMAR equivalent. Nevertheless, despite worse vision preoperatively, our results also showed that the mean postoperative BCVA was significantly better compared to baseline, with 73.3% of eyes exhibiting an improvement in visual acuity after surgery. This percentage is higher than what has been found in the literature, where only 40–66% of nanophthalmic eyes without glaucoma demonstrated improved visual acuity after routine cataract surgery [30,31]. Given the visual acuity benefit as well as refractive compensation for postoperative hyperopia, we would recommend primary implantation of IOLs in nanophthalmic glaucoma eyes at the time of surgery.

Not only does LPPV + ACSP + IOL + PC appear to be effective in lowering IOP and improving vision, but it is also safe for nanophthalmic glaucoma. Cataract surgery of nanophthalmic eyes is usually associated with a high risk of severe perioperative complications, such as choroidal effusions. Uveal effusions often occur following intraocular surgery because of sudden change in IOP and ACD and are a major cause of severe vision loss in nanophthalmic eyes. Yalvac et al. [11] evaluated the surgical results of trabeculectomy in patients with nanophthalmic glaucoma and found that choroidal detachments occurred in 50% and 25% of patients in the early and late postoperative periods, respectively. However, in our study, only 2 eyes developed localized choroidal detachments perioperatively, both of which resolved with conservative therapy. Neither early nor later uveal effusion, suprachoroidal hemorrhage, retinal detachment, or endophthalmitis was observed in the follow-up period. One of the most important reasons for the low incidence of complications was the surgical technique. Control of IOP fluctuations and stabilization of the anterior chamber were considered critical throughout the entire operation. To achieve this, LPPV was performed early to moderately decrease vitreous pressure, preventing acute IOP drops. Moreover, to avoid the dramatic pressure and ACD changes that occur from the sudden cessation of irrigation, viscoelastic agents were injected into the anterior chamber each time before the removal of the phacoemulsification or irrigation-aspiration tips from the eye. This step ensures the maintenance of IOP and ACD and prevents the development of severe uveal effusions or even expulsive choroidal hemorrhages. Chalam et al. [26] also demonstrated that anterior chamber maintainer may be another option to achieve anterior chamber stability during the phacoemulsification phase.

Day et al. [24] found that the occurrence of malignant glaucoma was 9.5% in nanophthalmic patients after cataract surgery; however, there were no cases of malignant glaucoma in our case series. This is likely because a posterior capsulotomy was performed during surgery, which is known to be effective for preventing aqueous misdirection [32]. While some anterior segment surgeons will create a posterior capsulotomy from an anterior approach through a corneal incision [32,33], we describe here and recommend using the 25-G vitrector through a scleral incision to form the posterior capsulotomy. The advantages of a posterior approach are manifold: first, the IOL can be kept in its original position without disturbing any zonules or the capsular bag; second, the surgeon is better able to control the size of the capsulotomy using the vitrector; and finally, the anterior vitreous attached to the posterior capsule can be simultaneously removed, decreasing the risk of postoperative aqueous misdirection.

Studies have reported that the risk for complications after incisional surgery is greater in eyes with nanophthalmic glaucoma [8,12]. However, in our study, no intra- or post-operative complications were observed in the 3 eyes that required Ahmed glaucoma valve implantation for additional IOP control. We suspect this is attributable to the effective decrease of positive vitreous pressure and increase of ACD achieved as a result of the modified procedure.

This study has several limitations. First, the retrospective nature may result in potential bias in our conclusions. Second, because this is a descriptive case series, the study did not include a control group. Finally, because of the short follow-up period, there may be long-term complications that are not captured in this study.

In summary, our study showed that altogether, LPPV + ACSP + IOL + PC is an effective and safe option for patients with complex nanophthalmos with successful IOP control, deepening of the ACD, improvement in visual acuity, and reduction of glaucoma medications required. Thus, this procedure may be considered for the treatment of eyes with complex nanophthalmos.

## Figures and Tables

**Figure 1 jcm-11-05909-f001:**
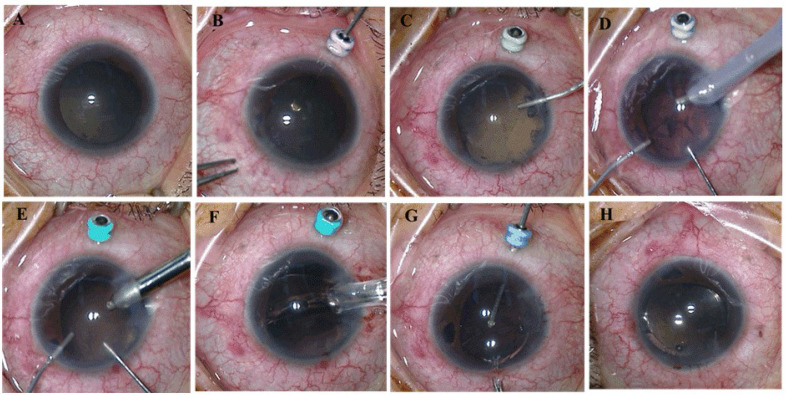
Surgical view of phacoemulsification combined with pars plana vitrectomy and posterior capsulotomy. (**A**) Preoperative view of the anterior segment. (**B**) Limited pars plana vitrectomy. A 25-gauge trocar was inserted in a transconjunctival and transscleral fashion 2 mm posterior to the limbus in the inferotemporal quadrant. (**C**) Deepening of the anterior chamber with a viscoelastic agent through a temporal clear corneal incision. (**D**,**E**) Prior to removal of the intraocular instruments (phaco or I/A tips), viscoelastic was injected into the anterior chamber from one paracentesis to prevent sudden change in IOP and ACD. (**F**) Implantation of a foldable single-piece IOL. (**G**) Posterior capsulotomy; (**H**) Removal of trocar.

**Figure 2 jcm-11-05909-f002:**
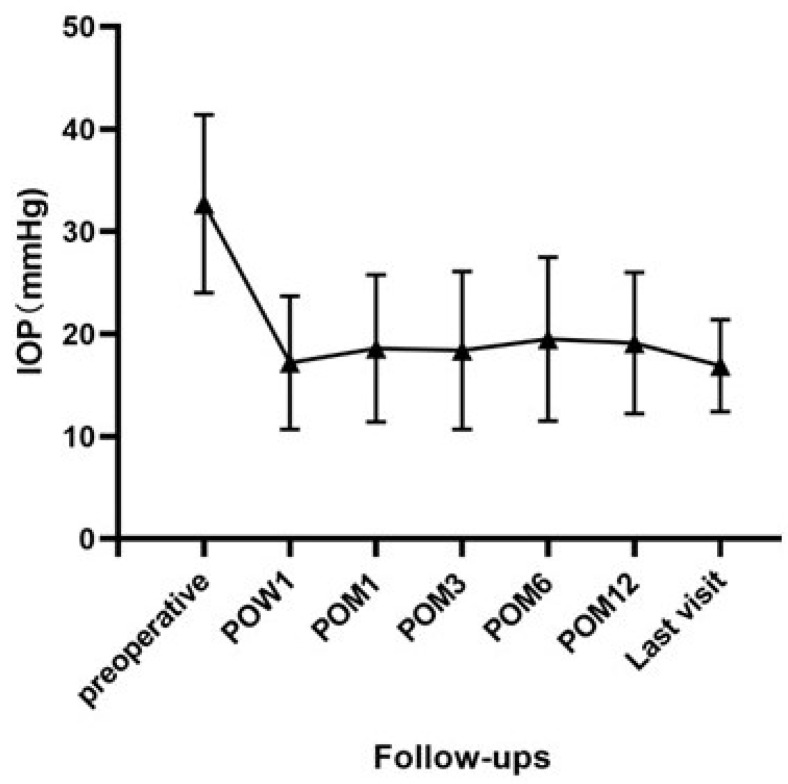
Pre- and postoperative intraocular pressure (IOP) changes at week 1 (POW1), month 1 (POM1), month 3 (POM3), month 6 (POM6), month 12 (POM12) and the final visit. The mean postoperative IOP was significantly reduced at all postoperative follow-up visits compared with preoperative measurements.

**Figure 3 jcm-11-05909-f003:**
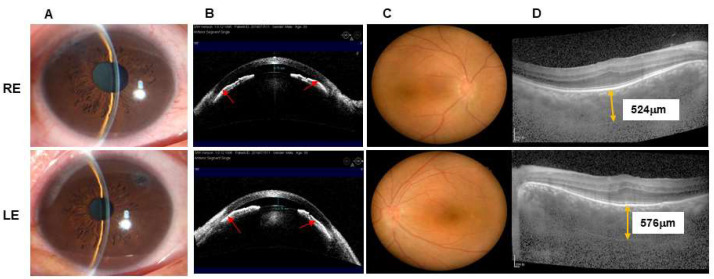
Preoperative images of 39-year-old man with nanophthalmos. (**A**,**B**) the preoperative anterior segment of both eyes showed a shallow anterior chamber (RE: 0.75 mm, LE: 0.76 mm) with angle closure (red arrow). (**C**) The fundus photographs of both eyes showed disappearance of macular fovea reflection. (**D**) Linear horizontal macular OCT scan revealed absence of a foveal depression, persistence of inner nuclear layers and thickened choroid. RE: right eye, LE: left eye.

**Figure 4 jcm-11-05909-f004:**
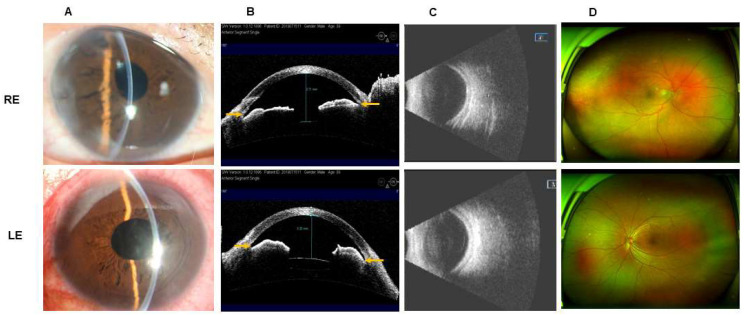
One-day postoperative images of the patient in Figure 1. (**A**,**B**) Postsurgical anterior segment of both eyes showed significantly increased center ACD (RE: 3.71 mm, LE: 3.33 mm) and open angle (yellow arrow). (**C**,**D**) B-scan ultrasonography and scanning laser ophthalmoscope showed flat retina in both eyes. RE: right eye, LE: left eye.

**Table 1 jcm-11-05909-t001:** Baseline characteristics of 37 patients with complex nanophthalmos (n = 45).

Characteristics	Mean ± SD/Median (Q1, Q3)	Range
IOP (mmHg)	32.7 ± 8.7	22.0–52.0
BCVA, logMAR	1.28 ± 0.64	0.2–2.6
HCD (mm)	10.7 ± 0.6	9.1–11.8
ACD (mm)	1.14 ± 0.51	0–2.22
LT (mm)	5.03 ± 0.75	3.08–5.99
AL (mm)	16.68 ± 1.18	14.67–19.50
LT/AL ratio (%)	30.4 ± 5.1	17.7–38.0
Degrees of angle closure ^&^	330 (210, 360)	0–360
RNFL thickness (μm) *	115.1 ± 52.8	24–237
SFCT (μm) *	472.3 ± 104.5	243.0–676.0
FRT (μm) *	350.7 ± 130.7	150.0–617.5
Number of medications	3 (2, 3)	1–4

IOP: intraocular pressure, BCVA: best-corrected visual acuity, HCD: horizontal corneal diameter, ACD: anterior angle chamber depth, LT: lens thickness, AL: axial length, RNFL: retinal nerve fiber layer, SFCT: subfoveal choroidal thickness, FRT: foveal retinal the thickness. Q1: 25th percentile, Q3: 75th percentile. ^&^: Ten eyes did not have gonioscopic exam due to corneal edema. * Three eyes were not able to obtain macular scan.

## Data Availability

Not applicable.

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
