# Peer review of "Effect of Combined Surgery in Patients with Complex Nanophthalmos"

_jcm, 2022, doi:10.3390/jcm11195909_

Round 1
Reviewer 1 Report
The manuscript title “Effect of combined surgery in patients with complex nanoph- 1 Thalmos” submitted to the Journal of Clinical Medicine by Wei et al. describes the evaluation of combined surgery for limited pars plana vitrectomy, anterior-chamber stabilized phacoemulsification, IOL implantation, and posterior capsulotomy, LPPV+ACSP+IOL+PC) in complex nanophthalmos, appeared to be exciting.
However, such surgeries have been reported by a couple of eye surgeons previously. Therefore, it could be better if the investigators provide a comprehensive review of the previously published report and clearly state how their techniques are superior to the existing ones.
The second point is that investigators might consider extending the follow-up duration beyond one year to confirm any complexity of the evolve in due course post-surgery. Overall, the manuscript is written very well and reads very well, and I believe it would be able to provide value addition to ophthalmological surgery.
Reviewer 2 Report
This manuscript reports the application of combined procedures in the cataract surgery for patients with nanophathalmos. The reported approach reduces complications and improve vision of most patients after 1 year follow-up. The authors have clearly addressed their surgical procedures, a typical case, and postoperative results. The manuscript can be more attractive to readers by changing a few confusing expressions and typos.
1. Line 56: “cataract who…” The clause “who desire…” should follow “patients”.
2. Line 62: “such as phacoemulsification alone, documented …”
3. Line 91: “The medical records of the enrolled patients were recorded.”
4. Line 237: The parentheses are in full width.
5. Line 289: “phacoemulsification, as it was previously performed for nanohthalmic glaucoma, has a few chanlleges”.
Reviewer 3 Report
Well-conducted study with adequate numerosity; critical points of surgery on complex nanophthalms were clearly identified; rationale for the surgical procedures adopted was presented.
Only a few clarifications are needed:
- line 369: "retrospective nature": It is also appropriate to specify in the abstract or in materials and methods the retrospective nature of the work.
- line 186: Median (Q1,Q3): Is unclear in the legend
- line 194: "at a week..": it's better to add "after surgery"
Reviewer 4 Report
- Congratulations to the authors for the article and the success in these complex surgeries.
- The manuscript is appropriately referenced and authors presented sufficient data with appropriate figures and the article is easy to read and logically structured
- Page 1 , Abstract “and reduce the number of glaucoma medications in patients with complex nanophthalmic.” . Please correct to nanophthalmos.
- You mentioned in methods section that in limited pars plana vitrectomy, trocar was placed 2mm posterior to limbus. Is this the same for all cases or guided by UBM or ultrasound?
